# MindVL: Towards Efficient and Effective Training of Multimodal Large Language Models on Ascend NPUs

## Abstract

We propose **MindVL**, a multimodal large language model (MLLMs) trained on Ascend NPUs. The training of state-of-the-art MLLMs is often confined to a limited set of hardware platforms and relies heavily on massive, undisclosed data recipes, which hinders reproducibility and open research. To change the common perception that Ascend hardware is unsuitable for efficient full-stage MLLM training, we introduce **MindSpeed-MLLM**, a highly efficient training framework that supports stable and high-performance training of large-scale Dense and Mixture-of-Experts (MoE) models on Ascend hardware. Based on this, we provide a systematic and open description of the data production methods and mixing strategies for all training stages. Furthermore, we present MindVL, a data-efficient multimodal large language model trained end-to-end on Ascend NPUs. In addition, we find that averaging weights from checkpoints trained with different sequence lengths is particularly effective and yields further gains when combined with test-time resolution search. Our experiments demonstrate superior data efficiency: **MindVL-8B** matches the performance of Qwen2.5VL-7B using only 10% of its training data, while our MoE model, **MindVL-671B-A37B**, matches Qwen2.5VL-72B using only 3% of the Qwen2.5VL training data, and achieves comparable performance with other leading multimodal MoE models. Our work provides the community with a valuable hardware alternative, open data recipes, and effective performance-enhancing techniques.

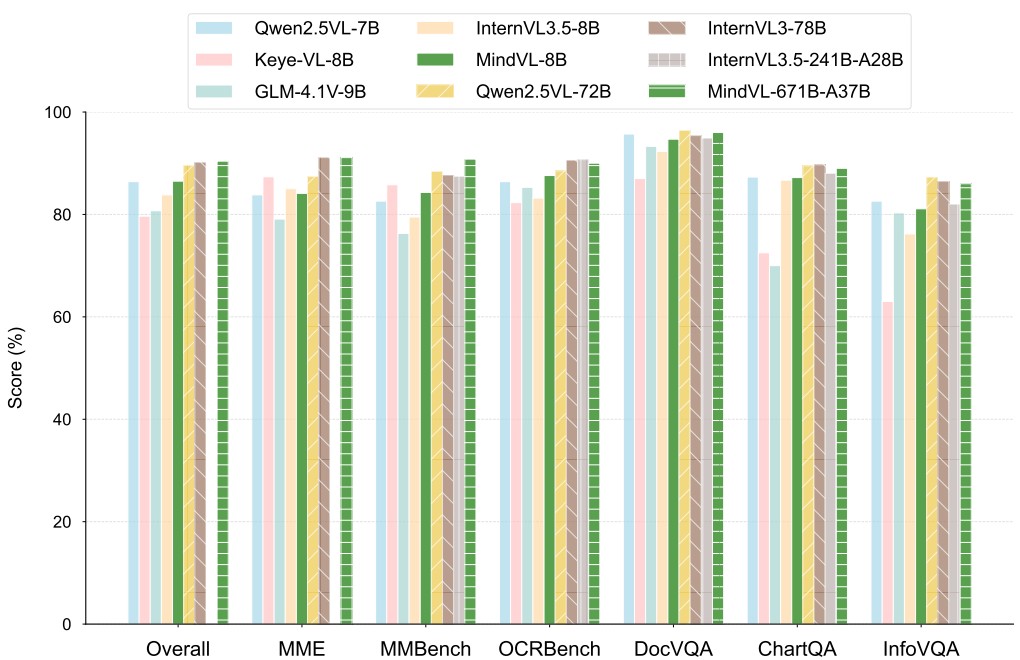

Figure 1: Benchmark performance of MindVL and its counterparts.

# 1 INTRODUCTION

Multimodal Large Language Models (MLLMs) (Chen et al., 2023a; Zhang et al., 2024a) represent a significant advancement in artificial intelligence, demonstrating remarkable capabilities in understanding and generating content across vision and language modalities. Despite rapid progress, the field faces two major challenges that limit open research. First, the training of top-tier models (Chen et al., 2023b; Guo et al., 2025; Chen et al., 2024; Bai et al., 2025; Team et al., 2025) is predominantly dependent on a specific hardware ecosystem (e.g., NVIDIA GPUs), creating a perception that alternative platforms like Ascend are incapable of efficient full-stage MLLM training. This perception restricts hardware choices for researchers. Second, while the composition of training data is widely acknowledged as a critical factor for performance, most leading models (Bai et al., 2025; Guo et al., 2025) only offer high-level descriptions of their data; the exact recipes, cleaning pipelines, and mixing strategies are often treated as proprietary secrets. This lack of transparency severely impedes reproducibility and hinders community progress.

To address these challenges, this work advocates for a more open and efficient research paradigm for MLLMs. Our primary contribution is **MindSpeed-MLLM**, an optimized training framework that demonstrates the full capability of Ascend hardware for stable and efficient training of both dense and large-scale Mixture-of-Experts (MoE) models from pre-training to supervised fine-tuning (SFT). This provides researchers with a crucial and performant hardware alternative.

Second, we aim to demystify the "black box" of leading MLLM data. We provide a comprehensive and open description of our data production methodology, including detailed data collection, cleaning, processing pipelines, and—most importantly—the mixing ratios used for each training stage. We believe this detailed data recipe offers a valuable blueprint for the community.

Third, we present **MindVL**, a data-efficient multimodal large language model trained end-to-end on Ascend NPUs. MindVL undergoes a three-phase training pipeline: warm-up, multitask training, and supervised instruction tuning, to incrementally enhance its multimodal capabilities. Starting with basic visual and cross-modal pre-training, the pipeline progresses to large-scale instruction adjustment, aligning the model with real-world use cases. Additionally, we integrate multimodal data packaging and hybrid parallelism techniques to significantly boost end-to-end training speed. To further optimize performance, we introduce two key strategies: test-time resolution search (to dynamically select optimal image resolutions for inference) and model weight averaging (to stabilize and improve final performance).

Extensive experiments validate the effectiveness of our approach. Our models achieve performance comparable to state-of-the-art models (e.g., the Qwen2.5VL series (Bai et al., 2025)) while utilizing orders of magnitude less training data. This result underscores the high quality of our data recipe and the robust training capability of the MindSpeed-MLLM framework on Ascend hardware. The contributions of this paper are as follows:

- We introduce **MindSpeed-MLLM**, a framework that enables efficient full-stage training of both Dense and MoE MLLMs on Ascend hardware, challenging existing perceptions.

- We provide a detailed and open data recipe for all training stages, promoting transparency and reproducibility in MLLM research.

- We introduce two enhancement techniques: multimodal model weight averaging and test-time resolution search, both of which contribute to improved pure text and multimodal performance.

- Experimental results show that MindVL achieves performance comparable to state-of-the-art models. Specifically, **MindVL-8B** matches the performance of Qwen2.5VL-7B using only 10% of its training data, while **MindVL-671B-A37B**, matches Qwen2.5VL-72B using only 3% of the Qwen2.5VL training data and and achieves comparable performance with other leading multimodal MoE models..

# 2 RELATED WORK

## 2.1 TRAINING OF MULTIMODAL LARGE LANGUAGE MODELS

The training of Multimodal Large Language Models (MLLMs) faces significant challenges due to model heterogeneity (e.g., integrating vision encoders with LLMs) and data heterogeneity (e.g., pro-

cessing images, videos, and text). These challenges necessitate specialized training frameworks and hardware optimizations to achieve efficiency and scalability. MLLM training relied on adapting text-centric frameworks like Megatron-LM (Shoeybi et al., 2019) and DeepSpeed (Rasley et al., 2020) to handle multimodal data by treating visual modules as additional layers within a unified parallelism strategy (e.g., combining Tensor, Pipeline, and Data Parallelism). Although numerous studies have successfully trained state-of-the-art MLLMs using these frameworks, their development has primarily focused on optimization for NVIDIA GPUs. In contrast, exploration of multimodal large-scale model training on Ascend NPUs remains limited, and a comprehensive, full-stage methodology for such environments has yet to be established.

## 2.2 Data Curation of Multimodal Large Language Models

Data curation is a cornerstone of multimodal large language model (MLLM) performance, as high-quality, well-structured multimodal data directly enables effective vision-language alignment and task adaptability. Existing literature and MLLM technical reports generally acknowledge the significance of data curation, outlining broad frameworks that typically categorize data by task and emphasizing core curation goals. Leading MLLMs, including Qwen2.5-VL (Bai et al., 2025) and Seed-VL 1.5 (Guo et al., 2025), provide only generalized descriptions of their data curation pipelines, lacking critical granular details that are essential for reproducibility and comparative analysis. This gap hinders the research community from fully dissecting how data curation choices impact MLLM capabilities and limits the development of future MLLMs.

## 3 MindSpeed-MLLM: Training Infrastructure on Ascend NPUs

Due to substantial hardware and software discrepancies, training frameworks widely used on NVIDIA GPUs—such as Megatron-LM (Shoeybi et al., 2019)—and common acceleration libraries (e.g., FlashAttention (Dao et al., 2022; Dao, 2024), Transformer-Engine (NVIDIA, 2023)) cannot be directly deployed on Ascend devices. [1] Thus, developing a robust distributed training framework for the Ascend ecosystem is essential. To this end, we introduce MindSpeed-MLLM: a distributed multimodal training library tailored for Ascend NPUs.

### 3.1 MindSpeed-MLLM

#### 3.1.1 MindSpeed Series Libraries

MindSpeed is a high-performance acceleration library tailored for the Ascend platform, encompassing three core components to support large-model training: MindSpeed-Core (Ascend, 2023), MindSpeed-LLM (Ascend, 2024a) MindSpeed-MM (Ascend, 2024b) and MindSpeed-RL (Feng et al., 2025).

MindSpeed-Core, built on and optimized for Ascend hardware based on Megatron-LM, delivers multi-dimensional optimizations in computing, memory, communication, and parallelism, enabling accelerated training for scenarios like long sequences and MoE. MindSpeed-LLM provides a rich set of LLM with extensive training optimization features, while MindSpeed-MM realizes mainstream Vision-Language Models.

Despite their individual strengths, the existing components present integration and functionality gaps for end-to-end multi-modal large language model (MLLM) training. Specifically, MindSpeed-MM lacks robust support for critical multi-modal data processing functionalities, including distributed data loading, data packing, and training resumption. Furthermore, its optimizations for the language backbone are not as comprehensive or mature as those provided by MindSpeed-LLM. To address these gaps and leverage the strengths of existing components, MindSpeed-MLLM is developed with targeted optimizations.

#### 3.1.2 MindSpeed-MLLM Framework

As depicted in Figure 2, the MindSpeed-MLLM framework is constructed with a hierarchical architecture. It builds upon the foundational optimizations from MindSpeed-Core and integrates partial modules from MindSpeed-LLM and MindSpeed-MM. Beyond this integration, the core efforts of MindSpeed-MLLM lie in enhancing multi-modal data processing, operator fusion replacement, and system-level scheduling optimization, which will be introduced separately in the following sections.

---

[1] A detailed analysis of NVIDIA-Ascend hardware and software differences is provided in Appendix A.

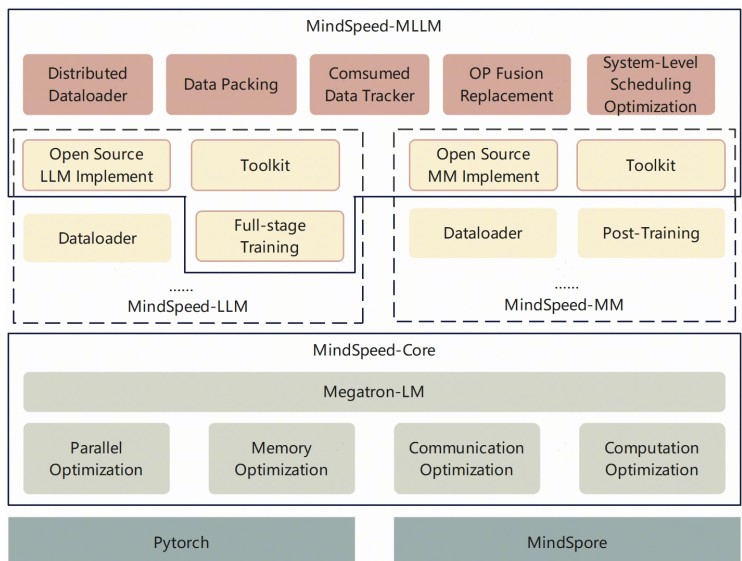

Figure 2: The Overall Architecture of MindSpeed-MLLM and Its Relationship with Other Mind-Speed Frameworks.

### 3.1.3 MULTI-MODAL DATA LOADER

We have developed a multi-modal data loader with the following features:

**Distributed Multi-Modal Data Loader:** It supports distributed data loading, where each data parallel group only reads the data within that group, effectively avoiding bottlenecks caused by reading the same data during large-scale training redundantly.

**Online Packing:** It enables online packing of multi-modal data, which combines data of different lengths into a specified length and fills in valid data content as much as possible Ma et al. (2025); Wang et al. (2025a). Thus, each Pack dataset has almost the same length, reducing the number of samples during training and improving training efficiency. Meanwhile, it controls the number of visual tokens to avoid load imbalance between pipeline parallel stages caused by uneven quantities of different modal data.

**Consumed Data Tracker:** It supports tracking of consumed data, which facilitates the location of data breakpoints during checkpoint-based recovery training, eliminates redundant data retraining, and ensures accurate resumption of training tasks.

### 3.1.4 OPERATOR FUSION REPLACEMENT

Choosing hardware-friendly operators effectively boosts training efficiency in model development. MindSpeed already supports fusion operator replacements, such as RMSNorm, SoftMax, MoE Token Permute, Unpermute, and Adamw, etc. Beyond these pre-provided fusion operators, MindSpeed-MLLM further optimizes as follows:

**Common Attention Patch:** MindSpeed-LLM and MindSpeed-MM replaced attention operators for language and visual components respectively, swapping the flash attention interface with npu fusion attention. Since both components require this operator, we patched attention in the shared transformer block to a unified fusion operator call. Different mask types are passed to specific components depending on the module ids.

**Mask Compression:** When training native resolutions ViTs, we first concatenate the hidden states of distinct images along the sequence dimension, followed by invoking the attention operator through a variable-length way. This design simultaneously reduces memory overhead and enhances computational efficiency. Combined with the sparse mode parameter of the npu fusion attention operator, passing the compressed attention mask further reduces the memory usage.

**Operation Replacement:** Profiling showed low computational efficiency of Conv operators in CANN 8.0. Thus, we replaced Conv2d and Conv3d operations with Matmul operation equivalently. The supporting checkpoint convert toolkit also added corresponding support for this replacement.

### 3.1.5 SYSTEM-LEVEL SCHEDULING OPTIMIZATION

Within the Ascend ecosystem, system-level scheduling optimizations deliver performance enhancements through multiple mechanisms. Fine-grained core binding minimizes cross-NUMA node memory accesses, reducing both task scheduling overhead and inter-core switching costs. Concurrently, operator deployment queue optimizations partition deployment tasks across multi-stage pipelines operating in parallel. This approach enables partial overlap between execution and submission processes, reducing overall latency and improving end-to-end performance.

## 4 DATA CURATION

The MindVL training corpus contains 447 billion diverse and high-quality tokens used for three training stages. The data is categorized according to target capabilities, and the curation process for each category is detailed in the following subsections. Due to space limitations, more detailed processing steps and data ratios are provided in the Appendix B.

As shown in Figure 3, the training data of MindVL is open-sourced into two main categories: image-text pairs and visual instructions. The image-text pair data is further divided into eight subcategories. Figure 3 illustrates the core processing methods for each category, along with fundamental filtering techniques and some models used during data annotation.

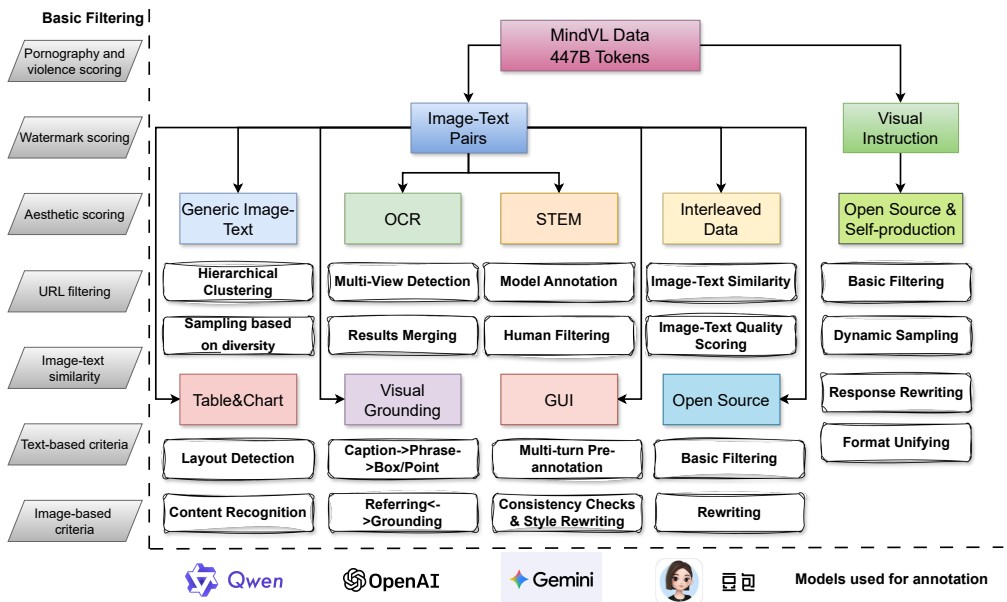

Figure 3: Data curation process of MindVL training data.

### 4.1 WARM-UP DATA

The MindVL warm-up corpus consists of 256 billion diverse, high-quality tokens. Warm-up data span six categories, including image caption, OCR, Visual Grounding, Table&Chart, GUI, STEM, with key processing steps described in Appendix B.1.

### 4.2 MULTITASK TRAINING DATA

Multitask training data consists of interleaved image-text (Table 12), visual instruction (Table 14, about 80B tokens), web2code (Yun et al., 2024) and text instruction (Table 13), totally 179B tokens. We add textual data to maintain the MindVL's linguistic capabilities. Details are described in Appendix B.2

### 4.3 SUPERVISED FINE-TUNING DATA

High-quality instruction data is sampled from open-source multitask datasets (classified by model), with low-quality answers re-annotated (via model and human verification), totally 12B tokens. Tex-

tual data is incorporated at a multimodal-to-language ratio of 1:1 to preserve the MLLM's linguistic performance. Details are described in Appendix B.3

# 5 MINDVL

## 5.1 ARCHITECTURE

The architecture of MindVL bears resemblance to that of Qwen2.5-VL, comprising three core components: a vision encoder, an MLP projector, and a large language model. The vision encoder natively supports dynamic image resolutions and adopts 2D RoPE (Wang et al., 2024b) for positional encoding, enabling flexible adaptation to images of arbitrary dimensions.

For model initialization, we utilize Qwen2.5ViT [2] as visual encoder and Qwen3 LLM (Yang et al., 2025) / DeepSeek-V3-0324 (Liu et al., 2024a) as language backbones, choices that ensure robust baseline performance. In contrast, the MLP projector is initialized randomly.

## 5.2 TRAINING RECIPE

As shown in Table 1, the training process of MindVL is divided into three distinct phases, each employing different data configurations and training strategies to progressively enhance the model's capabilities.

Table 1: Training setup and hyper-parameters in three training stages.

| Stages | Warm-up 8B / 671B | Multitask training 8B / 671B | SFT 8B / 671B |
|---|---|---|---|
| Training budget (tokens) | 256B/16B | 179B/80B | 12B |
| Sequence length | 8192 | 8192 | 2048/4096/8192 |
| Trainable components | MLP adaptor | All | All |
| Batch sizes | 1024 | 1024 | 512 |
| LR warm-up ratio | 0.1 | 0.1 | 0.1 |
| Maximum LR | 1e-3/2e-4 | 2e-5/5e-6 | 1e-5/2e-6 |
| Minimum LR | 0/1e-5 | 1e-5/5e-7 | 0/2e-7 |

During the warm-up phase, only the MLP adapter is trained to align the vision transformer and the language model. This process relies on carefully curated data spanning image captions, visual grounding, OCR, and STEM content to establish foundational multimodal alignment.

In the multitask training phase, all parameters are optimized with diverse multimodal data such as interleaved image-text, VQA, math reasoning, agent tasks, video, and pure text. This enhances complex visual-language reasoning while preserving linguistic abilities.

Finally, supervised fine-tuning unlocks all parameters and focuses on instruction-aware optimization to adapt pretrained representations to downstream tasks.

## 5.3 MODEL MERGING WITH DIFFERENT TRAINING SEQUENCE LENGTHS

Merging weights of deep models (Li et al., 2023) has been verified to be an efficient way in various applications, including multi-task learning, federated learning, model compression, and continual learning. We also explore the weight averaging strategies with the models trained under different settings of input sequence length and image resolution. Specifically, for the SFT phase, we train the model with sequence length of 2K, 4K and 8K with max_pixels of 1280*28*28, 3072*28*28 and 4096*28*28 respectively to enhance the model's adaptability to different context inputs and image resolutions. Finally, the MindVL model is created by averaging weights of models with different training sequence length.

## 5.4 TEST-TIME RESOLUTION SEARCH

The evaluation model is merged by model weights trained with different sequence lengths as well as different max_pixels threshold of training images. Such search strategy is generally effective as the original resolution of the test images maybe be out of distribution relative to that of the training images. Therefore, we conduct a grid search about up-scaling small images to surpass a specified min_pixels threshold and down-scaling high-resolution image to be lower than a specified

---

[2]The weights of Qwen2.5ViT are derived from the Qwen2.5-VL 72B model.

max_pixels threshold. Specifically, we set the grid search space of the min_pixels as {4, 16, 32, 64}*28*28 and max_pixels as {1280, 2048, 2560,3072,4096,8192}*28 *28. The analysis results are presented in Section 6.3.

# 6 EXPERIMENTS

## 6.1 PERFORMANCE OF MINDVL-8B

As shown in Table 2, we evaluate the overall performance of MindVL-8B on MMBench (Liu et al., 2024c), MME (Chaoyou et al., 2023), OCRBench (Liu et al., 2024d), DocVQA (Mathew et al., 2021), ChartQA (Masry et al., 2022), InfoVQA (Mathew et al., 2022). Overall, MindVL-8B outperforms several leading models, including Qwen2.5-VL-7B, GLM-4.1V-9B, Keya-VL-8B, and InternVL3.5-8B. Notably, MindVL-8B achieves this superior performance using only 447B tokens of training data—roughly one-tenth of the data used by Qwen2.5-VL-7B. Furthermore, compared to models trained with trillion-scale tokens such as GLM-4.1V-9B and Keye-VL-8B, MindVL delivers significantly stronger results, outperforming them by margins of 6.6 and 5.8 points, respectively. When compared to InternVL3.5-8B, which has a similar pre-training scale, MindVL-8B maintains a lead of 2.7 points. These outcomes highlight the effectiveness of our training methodology and data efficiency, demonstrating the capability to develop high-performing multimodal large language models on Ascend NPUs.

Table 2: Performance of MindVL-8B and comparison models on multimodal benchmarks. "+" indicates that there is a portion of data with unlabeled quantities. The results of the comparative models are referenced from (Wang et al., 2025b).

| Model | #Tokens | MME | MMBench | OCRBench | DocVQA | ChartQA | InfoVQA | Overall |
|---|---|---|---|---|---|---|---|---|
| Qwen2.5-VL-7B (Bai et al., 2025) | 4.1T+ | 83.8 | 82.6 | 86.4 | 95.7 | 87.3 | 82.6 | 86.4 |
| Keye-VL-8B (Team, 2025) | 1T+ | 87.3 | 85.8 | 82.3 | 87.0 | 72.5 | 63.0 | 79.9 |
| GLM-4.1V-9B (Team et al., 2025) | 2T+ | 79.1 | 76.3 | 85.3 | 93.3 | 70.0 | 80.3 | 80.7 |
| InternVL3.5-8B (Wang et al., 2025b) | 380B+ | 85.0 | 79.5 | 83.2 | 92.3 | 86.7 | 76.2 | 83.8 |
| MindVL-8B | 447B | 84.1 | 84.3 | 87.6 | 94.7 | 87.2 | 81.1 | 86.5 |

## 6.2 PERFORMANCE OF MINDVL-671B-A37B

As shown in Table 3, we compare our MindVL-671B-A37B with open source models of approximate parameter size and closed source excellent models on OCR, chart, and document understanding benchmarks. Overall, our MindVL-671B-A37B, trained with 106B multimodal tokens, achieves best average score across above benchmarks. These results show that our data and training recipes are effective in Vision-Language Alignment for large language models with different parameter scales, i.e., Qwen3-8B and DeepSeek-V3.

In addition, the comparison results of MindVL-671B-A37B and other models on multi-modal Hallusion and STEM benchmarks, including HallusionBench (Guan et al., 2024), AI2D (Kembhavi et al., 2016), MMVet (Yu et al., 2023), MathVista (Lu et al., 2023), MathVision (Wang et al., 2024a) and MMMU pro (Yue et al., 2024), are shown in Table 4. MindVL-671B-A37B gets the best evaluation results on HallusionBench. Although MindVL-671B-A37B achieves competitive performance against Qwen2-VL-72B and Qwen2.5-VL-72B on overall score, some of the model's multimodal reasoning capabilities are not sufficiently activated. For example, there is still a certain gap between our model and the state-of-the-art models on MMVet, MathVision and MMMU pro. The reason is that MindVL-671B-A37B is trained with only 106B multimodal tokens. In the future, we will add rich interleaved image-text data to boost model reasoning ability with broad domain knowledge.

Moreover, as shown in Table 5, we report the pure text evaluation results of MindVL-671B-A37B cross diverse language benchmarks, including AIME2024 (MAA, 2024), AIME2025 (MAA, 2025), GPQA-D (Rein et al., 2024), IFEval (Zhou et al., 2023), ArenaHard (Li et al., 2024), C-SimpleQA (He et al., 2024) and C-Eval (Huang et al., 2023). MindVL-671B-A37B outperforms the original DeepSeek-V3 in Overall evaluation result. Specifically, MindVL-671B-A37B achieves significant improvements on AIME2024, AIME2025, IFEval and AreanaHard datasets, and maintained the pure text capabilities of DeepSeek-V3 on GPQA-D, C-SimpleQA and C-Eval datasets. In addition, MindVL-671B-A37B also obtains better evaluation results than Qwen2.5VL-32B and very competitive results compared to Qwen2.5VL-72B across these pure text benchmarks.

Table 3: Comparison of OCR, chart, document and general understanding performance.

| Model | #Tokens | MME | MMBench | OCR Bench | DocVQA | ChartQA | InfoVQA | Overall |
|---|---|---|---|---|---|---|---|---|
| GPT-4V (OpenAI, 2023) | – | 68.8 | 80.0 | 64.5 | 88.4 | 78.5 | 75.1 | 70.0 |
| GPT-4o-20240513 (Hurst et al., 2024) | – | – | 83.1 | 73.6 | 92.8 | 85.7 | 79.2 | – |
| Claude-3-Opus (Anthropic, 2024a) | – | 56.7 | 60.1 | 69.4 | 89.3 | 80.8 | 55.6 | 67.3 |
| Claude-3.5-Sonnet (Anthropic, 2024b) | – | – | 80.9 | 78.8 | 95.2 | 90.8 | 74.3 | – |
| Gemini-1.5-Pro (Team et al., 2024) | – | – | 74.6 | 75.4 | 93.1 | 87.2 | 81.0 | – |
| Step3V (StepFun, 2025) | – | – | 81.1 | 83.7 | – | – | – | – |
| GLM-4.5V (Team et al., 2025) | 2T+ | – | 86.7 | 87.2 | 94.5 | 86.6 | 84.1 | 75.8 |
| Qwen2-VL-72B (Wang et al., 2024b) | 1.4T+ | 88.7 | 85.9 | 87.7 | **96.5** | 88.3 | 84.5 | 88.6 |
| Qwen2.5-VL-72B (Bai et al., 2025) | 4.1T+ | 87.4 | 88.4 | 88.5 | 96.4 | 89.5 | 87.3 | 89.6 |
| InternVL3-78B (Zhu et al., 2025) | 200B+ | **91.1** | 87.7 | 90.6 | 95.4 | **89.7** | 86.5 | 90.2 |
| InternVL3.5-241B-A28B (Wang et al., 2025b) | 380B+ | – | 87.4 | **90.7** | 94.9 | 88.0 | 82.0 | – |
| MindVL-671B-A37B | 106B | **91.1** | **90.8** | 90.0 | 96.0 | 89.0 | **88.9** | **91.0** |

Table 4: Comparison of multi-modal Hallusion/STEM performance.

| Model | #Tokens | Hallusion Bench | AI2D | MMVet | MathVista | MathVision | MMMU pro | Overall |
|---|---|---|---|---|---|---|---|---|
| GLM-4.5V (Team et al., 2025) | 2T+ | 65.4 | 86.6 | 75.2 | 78.2 | 52.5 | **59.8** | **69.6** |
| Qwen2-VL-72B (Wang et al., 2024b) | 1.4T+ | 58.1 | 88.1 | 74.0 | 70.5 | 25.9 | 46.2 | 60.5 |
| Qwen2.5-VL-72B (Bai et al., 2025) | 4.1T+ | 55.2 | 88.7 | 76.2 | 74.2 | 38.1 | 51.1 | 63.9 |
| InternVL3-78B (Zhu et al., 2025) | 200B+ | 59.1 | **89.7** | **81.3** | 79.0 | 43.1 | – | – |
| InternVL3.5-241B-A28B (Wang et al., 2025b) | 380B+ | 57.3 | 87.3 | 81.2 | **82.7** | **63.9** | – | – |
| MindVL-671B-A37B | 106B | **68.6** | 85.2 | 67.9 | 72.9 | 34.1 | 49.5 | 63.0 |

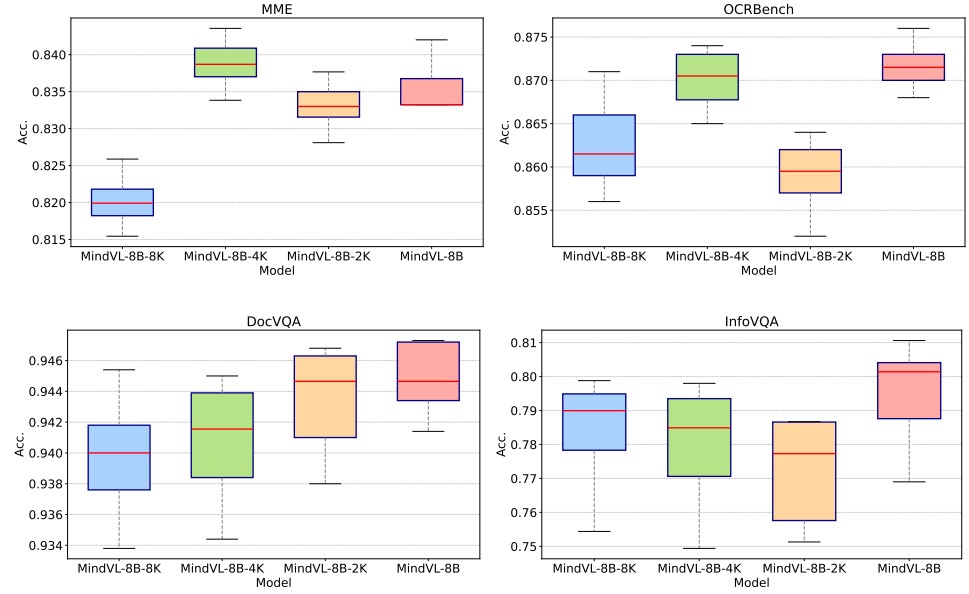

Figure 4: Box plots of accuracies with varying input images resolutions for different models. Maximum value, minimum value, median and quartiles are plotted.

## 6.3 EFFECTIVENESS OF MODEL MERGING WITH DIFFERENT TRAINING SEQUENCE LENGTHS

Table 6 validate the effectiveness of our model merging strategy, which is implemented by averaging the weights of models trained on sequence lengths of 2K, 4K, and 8K—denoted as MindVL-8B-2K, MindVL-8B-4K, and MindVL-8B-8K, respectively. The merged model is denoted as MindVL-8B. Our MindVL-8B model achieves an average benchmark performance of 86.5%, which are better than the scores of MindVL-8B-2K, MindVL-8B-4K, and MindVL-8B-8K. The proposed strategy significantly surpasses the perceptual capabilities of MindVL-8B, especially on MMBench, ChartQA and InfoVQA datasets.

Figures 4 shows the box plots of the evaluation results using test-time image resolution search strategy as mentioned in Section 5.4. On OCRBench, DocVQA and InfoVQA, MindVL-8B outperforms other models not only in terms of the highest accuracy scores across different image resolutions, but

Table 5: **Comparison of model performance across diverse language benchmarks.** All results are uniformly evaluated by the internal testing platform.

| Model | AIME2024 | AIME2025 | GPQA-D | IFEval | ArenaHard | C-SimpleQA | C-Eval | Overall |
|---|---|---|---|---|---|---|---|---|
| Qwen2.5VL-32B (Bai et al., 2025) | 30.0 | 16.7 | 51.5 | 64.3 | 92.2 | 44.9 | 81.3 | 54.1 |
| Qwen2.5VL-72B (Bai et al., 2025) | 26.7 | 16.7 | 50.0 | 85.6 | 71.2 | 49.8 | 86.1 | 55.2 |
| DeepSeek-V3-0324 (Liu et al., 2024a) | 60.0 | 43.3 | **69.2** | 81.9 | 94.8 | **73.3** | **89.6** | 73.2 |
| MindVL-671B-A37B | **63.3** | **46.7** | 68.7 | **84.7** | **97.0** | 72.9 | 88.2 | **74.5** |

Table 6: Studies of model merging strategy of MindVL-8B on multimodal benchmarks.

| Model | Sequence Length | Maximum Pixels | MME | MMBench | OCRBench | DocVQA | ChartQA | InfoVQA | Overall |
|---|---|---|---|---|---|---|---|---|---|
| MindVL-8B-2K | 2K | 1280 | 81.6 | 82.4 | 87.0 | 94.2 | 86.8 | 79.6 | 85.3 |
| MindVL-8B-4K | 4K | 3072 | **84.3** | 82.3 | 87.3 | 94.5 | 86.6 | 79.8 | 85.8 |
| MindVL-8B-8K | 8K | 4096 | 82.5 | 82.5 | 86.0 | 94.6 | 86.4 | 79.9 | 85.3 |
| MindVL-8B | - | - | 84.1 | **84.3** | **87.6** | **94.7** | **87.2** | **81.1** | **86.5** |

Table 7: Studies of model merging strategy of MindVL-671B-A37B on language benchmarks.

| Model | AIME2024 | AIME2025 | GPQA-D | IFEval | ArenaHard | C-SimpleQA | C-Eval | Overall |
|---|---|---|---|---|---|---|---|---|
| DeepSeek-V3-0324 (Liu et al., 2024a) | 60.0 | 43.3 | **69.2** | 81.9 | 94.8 | **73.3** | **89.6** | 73.2 |
| MindVL-671B-A37B-4K | 20.0 | 16.7 | 58.6 | 81.5 | 91.0 | 70.8 | 84.4 | 60.4 |
| MindVL-671B-A37B | **63.3** | **46.7** | 68.7 | **84.7** | **97.0** | 72.9 | 88.2 | **74.5** |

Table 8: Studies of model merging strategy of MindVL-671B-A37B on multimodal benchmarks. All results are uniformly evaluated by the internal testing platform (Correctness Evaluation).

| Model | MM-Bench | OCR-Bench | DocVQA Val | ChartQA | InfoVQA Val | AI2D | MMVet | Math-Vista | Math-Vision | MMMU pro | Overall |
|---|---|---|---|---|---|---|---|---|---|---|---|
| MindVL-671B-A37B-4K | **90.8** | **90.0** | **96.0** | **79.6** | 83.0 | **85.2** | 67.9 | **72.9** | 33.5 | 45.0 | 74.4 |
| MindVL-671B-A37B | 90.7 | 89.4 | 95.8 | 78.7 | **84.2** | 83.2 | **70.2** | 70.7 | **43.0** | **49.1** | **75.5** |

also shows improvements in both the lowest scores and median scores. On MME dataset, MindVL-8B outperforms MindVL-8B-2K and MindVL-8B-8K. The above results demonstrate that the model weight merging strategy can enhance the model's robustness against variations in input images of different resolutions. For specific numerical results, please refer to Appendix C.1.

In addition, we verified the effectiveness of the model merging strategy for large-scale language model, i.e. DeepSeek-V3, in maintaining the pure text ability after Vision-Language alignment. As shown in Table 7, MindVL-671B-A37B is obtained by merging the language models of DeepSeek-V3 and MindVL-671B-A37B-4K, i.e., the SFT model trained with 4K sequence length. MindVL-671B-A37B not only outperforms MindVL-671B-A37B-4K on each benchmark, but also outperforms the original language model DeepSeek-V3 on AIME2024, AIME2025, IFeval, ArenaHard and overall average score. For multimodal performance, MindVL-671B-A37B achieves the evaluation results similar to MindVL-671B-A37B-4K, as shown in Table 8.

# 7 CONCLUSION

In this paper, our proposed MindVL, trained on MindSpeed-MLLM framework with Ascend NPUs. We highlight the effectiveness of our data recipe and the robust training capability of the framework on Ascend hardware. Key contributions include the introduction of an efficient full-stage training framework for both Dense and MoE MLLMs, a transparent data recipe, and novel enhancement techniques. Notably, MindVL-8B and MindVL-671B-A37B achieve competitive results on both multimodal benchmark and language benchmark using only a fraction of the training data required by comparable models. Moreover, our proposed model merging method has effectively enhanced the model's performance. We find that model merging is an effective and low-cost approach to improve model performance. In future work, we will further explore the methodologies of model merging and the underlying principles behind it.

## 8 REPRODUCIBILITY STATEMENT

We have provided a detailed description in the main text and appendix of the training framework MindSpeed-MLLM (Section 3 and Appendix A) used by MindVL, the data and data ratios for each stage (Section 4 and Appendix B), and the training hyper-parameters of the model at each stage (Section 5), all to facilitate researchers in referencing and reproducing our work. Furthermore, following internal review, we will make the MindSpeed-MLLM code open-source to advance research on multimodal large language models on Ascend NPUs.

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
