# OpenReview forum: "MindVL: Towards Efficient and Effective Training of Multimodal Large Language Models on Ascend NPUs"
_ICLR.cc/2026/Conference — ICLR 2026 Conference Withdrawn Submission_

### Official Review · Reviewer_4Gna · 2025-10-28

**Soundness:** 1
**Presentation:** 2
**Contribution:** 1
**Rating:** 2
**Confidence:** 5

**Summary:**

This work presents MindVL, a competitive VLM which is trained with NPUs. The paper presents the data, infra, and training framework for developing this VLM. Additionally, some data statistics and framework features are included within the work.

Overall, you do not need to submit this technical report to ICLR conference. If you would like to make such paper submission, you may need to claim and present your technical contributions. Outstanding VLMs like InternVL and QwenVL are just released on arxiv instead of included in conference proceedings. If you are confident about your work, then publish in tech report to earn your credits. No need to submit it to the conference to earn credits which are not helpful in building your reputation.

To make a valid paper submission, you should at least specify the following details in VLM development: 1) detailed data curation pipeline, which data quality classifier is used, the prompts for model based data curation, the original sources for collecting the data, the human annotator rubric and criteria, the clustering embedding models and sampling algorithm; 2) the online packing implementations and algorithms in Section 3.1.3. Please consider to follow SAIL-VL and OpenQwen2VL to reposition your contributions.

**Strengths:**

Check summary.

**Weaknesses:**

Check summary.

**Questions:**

None

---

### Official Review · Reviewer_pUiZ · 2025-10-31

**Soundness:** 2
**Presentation:** 2
**Contribution:** 2
**Rating:** 2
**Confidence:** 4

**Summary:**

This paper proposes MindVL, a multimodal LLM trained on Ascend NPUs. To deal with specific challenges on Ascend hardware, they propose an optimized training framework named MindSpeed-MLLM. They provide a new recipe of data construction pipeline for multimodal-LLMs. The resulted model is named MindVL and achieve on-par performance with Qwen2.5-VL.

**Strengths:**

-	It seems to be a comprehensive work involving infrastructure, data curation, model tuning and implementation. The technical execution is solid, and the resulted model achieves good performance on multimodal understanding benchmarks.

**Weaknesses:**

-	The whole paper looks like a technical report and does not have a core technical contribution. For the training library, the authors provide explanation on data loader, operator fusion, system-level scheduling optimization. However, these explanations are all on high-level, and it seems there is nothing new compared to existing training frameworks. The authors did not explain clearly what part of the Ascend hardware needs special handling for the large-scale training.
-	The data curation part uses proprietary models (qwen, openAI, gemini, doubao) to annotate the data. It is used in multiple stages (filtering, annotation, rewrite), it is hard to know how much contribution of this external knowledge source has on the final performance.  Due to this, it is unfair to directly compare amount of training data with existing models (line 99-101).
-	Some technical details are not clear. 1. What does the author do specifically to support MoE models? 2. For mask compression (line 209-212), how is the memory overhead and computation efficiency improved?

**Questions:**

-	The authors mentioned the data will be open-sourced. However, given the scale of the data, how will it be open-sourced and made publicly accessible? (line 230)

---

### Official Review · Reviewer_gmxU · 2025-11-01

**Soundness:** 2
**Presentation:** 2
**Contribution:** 2
**Rating:** 2
**Confidence:** 4

**Summary:**

This paper introduces MindVL, a multimodal large language model trained on Ascend NPUs. Firstly, this work presents MindSpeed-MLLM, a highly efficient training library designed to support the training of MLLMs with both dense and MoE architectures on Ascend NPU hardware. Moreover, the paper introduces MindVL, a data-efficient MLLM, which is further enhanced through multimodal model weight averaging and test-time resolution search techniques. Experimental results demonstrate that MindVL achieves state-of-the-art performance while utilizing less training data.

**Strengths:**

- This paper provides a training framework that enables the training MLLMs on Ascend NPU hardware.
- The trained MindVL model achieves competitive performance compared to state-of-the-art MLLMs, while using less data.
- The training data recipe is open-sourced.

**Weaknesses:**

- Although the data recipe is public, the data itself is annotated using closed-source API models including gemini, gpt, and doubao. This process is costly, making it difficult for other researchers to replicate.

- For data construction, identifying low-quality data and performing re-annotation is crucial. However, the paper lacks a detailed explanation of how this process is specifically conducted.

- The test-time resolution search method lacks comparison with other models. Additionally, it's unclear whether this test-time method can be effectively transferred to other models.

**Questions:**

- Will the training data be open-sourced? While the data recipe is public, the actual data is annotated using closed-source models, making replication challenging for other researchers.

- Could you elaborate on the process for filtering low-quality data and re-annotating it? Specifically, how is the combination of human and model verification utilized? This is a crucial aspect that requires clarification.

---

### Official Review · Reviewer_KT7G · 2025-11-01

**Soundness:** 3
**Presentation:** 2
**Contribution:** 2
**Rating:** 4
**Confidence:** 3

**Summary:**

The paper MindVL introduces an efficient framework (MindSpeed-MLLM) for training multimodal large language models (MLLMs) on Ascend NPUs, challenging the dominance of NVIDIA GPUs. It provides open, reproducible data recipes and introduces two key techniques—model weight averaging and test-time resolution search—to enhance performance. Results show that MindVL-8B matches or exceeds Qwen2.5VL-7B performance while using 10% of its training data, and MindVL-671B-A37B matches Qwen2.5VL-72B performance with only 3% of the data. Both models perform strongly on multimodal benchmarks, demonstrating the viability of Ascend NPUs for large-scale MLLM training.

**Strengths:**

1. Efficiency and Transparency: MindVL achieves high performance with far less data while openly detailing its data recipes and training setup, improving reproducibility.

2. Hardware Innovation: It proves that Ascend NPUs can effectively train large multimodal models, expanding hardware options beyond NVIDIA GPUs.

3. Practical Enhancements: Novel methods like model weight averaging and test-time resolution search boost robustness and multimodal alignment at low computational cost.

**Weaknesses:**

1. Limited Architectural Novelty: The model mainly builds on existing designs like Qwen2.5-VL, offering engineering improvements rather than new modeling concepts.

2. Incomplete Reasoning Performance: MindVL underperforms top models on complex multimodal and STEM reasoning tasks.

3. Restricted Accessibility: Dependence on Ascend hardware limits reproducibility and adoption outside that ecosystem.

**Questions:**

The paper claims high training efficiency but doesn’t provide concrete hardware utilization statistics (e.g., throughput, FLOPs/s, energy use) to compare with NVIDIA baselines.

---

### Note · Authors · 2025-11-28

I have read and agree with the venue's withdrawal policy on behalf of myself and my co-authors.